# User-Centric Mixed Reality Interventions for Parkinson's Tremor Management: A Path Toward Digital Therapeutics

1st Zhenhong Lei*
*Architecture Department*
*Rhode Island School of Design*
Providence, US
brad.zh.lei@gmail.com
https://orcid.org/0009-0006-3734-7421

1st Xinjun Li*
*Information Science*
*Cornell University*
New York, US
xl863@cornell.edu
https://orcid.org/0009-0005-5171-3566

*Abstract*—**Mixed reality (MR) presents a promising frontier in rehabilitation sciences, particularly for addressing motor impairments associated with Parkinson's disease (PD). This paper explores the integration of an MR-based interactive rehabilitation system with an ergonomic hand-support device, designed to facilitate tremor stabilization and enhance fine motor control. By leveraging real-time sensor feedback and dynamic exercise progression, the system fosters an adaptive and engaging therapeutic environment. Conventional rehabilitation methods have demonstrated limited efficacy in addressing the progressive nature of PD-related tremors, with studies reporting that over 60% of patients experience difficulty in executing daily motor tasks. Through an iterative development process informed by expert consultation and patient feedback, the proposed approach emphasizes usability and clinical feasibility. Although preliminary in scope, this study provides a foundational framework for Mixed Reality-assisted rehabilitation and suggests future pathways for adaptive therapeutic interventions. The findings contribute to the growing discourse on digital health innovation, underscoring MR's potential to augment traditional rehabilitation strategies through immersive, user-centered methodologies.**

*Keywords*—**Augmented Reality, Mixed Reality, Hand Rehabilitation, Parkinson's Disease, Assistive Device, Haptic Feedback**

## I. INTRODUCTION

Parkinson's disease (PD) is a progressive neurodegenerative disorder that affects approximately 1% of individuals over 60 years of age, with prevalence increasing to nearly 4% in those over 80 [1]. Characterized by motor dysfunctions such as tremors, rigidity, and bradykinesia, PD significantly diminishes an individual's capacity for independent movement and fine motor tasks. Rehabilitation remains a cornerstone of PD management, yet conventional therapeutic modalities, including physiotherapy and mechanical assistive devices, often fall short in providing sustained engagement and adaptability to individualized patient needs [2]. Adherence rates to traditional rehabilitation programs remain low due to lack of motivation,

*Authors contributed equally and shall both be considered first authors

static therapeutic designs, and insufficient feedback mechanisms, necessitating the exploration of alternative solutions that offer both adaptability and sustained patient engagement. Mixed reality (MR) has emerged as a transformative tool in digital health, enabling immersive rehabilitation environments that provide real-time feedback and interactive task execution. Studies have shown that MR-assisted rehabilitation can yield up to a 30% improvement in motor function outcomes compared to conventional approaches, primarily by enhancing user engagement and neuroplasticity-driven motor learning. By leveraging precise motion tracking and adaptive task progression, MR facilitates the personalization of rehabilitation exercises, accommodating the unique motor profiles of PD patients. Despite these advantages, the application of MR in tremor rehabilitation remains underexplored, with existing research largely focusing on general motor rehabilitation rather than targeted interventions for movement disorders. This paper presents a preliminary exploration of a mixed-reality-assisted tremor rehabilitation system, integrating an ergonomic hand-support mechanism with interactive MR exercises to stabilize tremors and enhance fine motor skills. The system was developed through a rigorous design process incorporating feedback from healthcare professionals and individuals affected by tremor disorders, ensuring a balance between therapeutic efficacy and usability. By situating MR within the broader landscape of neurorehabilitation, this work aims to contribute to the evolving discourse on digital health interventions for movement disorders.

## II. METHODOLOGY

This study presents the development of a Mixed Reality-driven rehabilitation system designed to support tremor rehabilitation in PD patients by integrating physical stabilization with adaptive, MR-based motor training exercises. To inform the system design, we conducted a structured interview study with 28 participants, consisting of 6 healthcare professionals

(clinicians, rehabilitation specialists, and researchers) and 22 PD patients at varying disease stages. Table I & II provides a comprehensive overview of the participants' background information. The study aimed to address three key research questions: (1) identifying fundamental user needs in MR rehabilitation for tremor management, (2) understanding user perceptions of MR-based haptic and audiovisual feedback, and (3) investigating usability and accessibility challenges for home-based MR therapy.

### A. System Design and Thematic Insights

The development process began with a comparative analysis of existing rehabilitation approaches, assessing the strengths and limitations of mechanical assistive devices and virtual rehabilitation platforms. Mechanical solutions such as adaptive utensil aids and stabilizers provide immediate tremor support by counteracting involuntary movement, though their rehabilitative impact remains limited over time [3]. In contrast, MR-based rehabilitation utilizes immersive simulated activities to promote motor skill improvement but lacks direct physical support during real-world tasks [4]. The structured interviews reinforced the need for an integrated approach, combining physical ergonomic assistance with adaptive virtual MR therapy to enhance engagement, usability, and rehabilitative effectiveness. Findings from the interviews confirmed that healthcare professionals prioritize personalization and adaptability, with 83% of clinicians emphasizing that static rehabilitation tools fail to address individual differences in tremor severity and motor response. Additionally, 77% of PD patients expressed frustration with traditional rehabilitation methods, citing monotony, lack of engagement, and limited performance feedback. Among patients with prior exposure to VR or AI-driven therapy (N=7), interactive rehabilitation environments were perceived as more motivating. However, concerns related to usability barriers, including motion sickness (N=3) and cognitive load (N=5), were identified, indicating a need for an intuitive, user-friendly system design.

### B. Mixed Reality-Driven Adaptation and Haptic Feedback Integration

A critical limitation highlighted by 5 out of 6 healthcare professionals was the lack of real-time motor adaptation in existing MR-based rehabilitation solutions. Current fixed training modules do not adjust to individual variations in tremor patterns, requiring AI-based adaptive learning mechanisms to dynamically scale exercise difficulty, stabilization force, and feedback intensity based on sensor-tracked user performance [5]. To address this, the system integrates real-time motion tracking and tremor compensation algorithms, ensuring that rehabilitation tasks remain both challenging and achievable. Haptic feedback also emerged as an essential feature, with 68% of patients responding positively to haptic-assisted motor guidance, suggesting that physical reinforcement enhances proprioceptive awareness and motor confidence. Clinicians highlighted that haptic feedback mechanisms within MR environments can improve motor coordination and fine motor skill

retention, underscoring its role in enhancing task execution precision and training effectiveness over time [6].

### C. System Implementation: Physical and Virtual Components

The refined system architecture, informed by thematic analysis, incorporates both a physical ergonomic hand-support device and an MR-based adaptive training environment, bridging the gap between biomechanical support and interactive therapy.

- Physical Component: A custom-designed ergonomic hand motion assistance device provides real-time stabilization and force feedback, addressing clinician concerns regarding tremor-induced motor instability. This device integrates a servo-driven adaptive grip mechanism, which dynamically compensates for tremor oscillations [5].
- Virtual MR Component: The MR-based rehabilitation interface allows for real-time adjustments based on individual progress. Unlike static rehabilitation programs, this system incorporates progressive motor learning principles, ensuring that tasks remain appropriately challenging while facilitating gradual motor improvement.
- Haptic-Integrated Feedback Mechanism: The integration of force-feedback gloves and vibrotactile stimulation enhances proprioceptive input, reinforcing motor learning through physical guidance. This implementation was based on patient responses indicating that haptic guidance improves movement awareness and task execution precision, aligning with clinical best practices for neurorehabilitation [6].

### D. Evaluation and Future Considerations

To validate the system's effectiveness, structured user assessments were conducted, comparing the Mixed Reality-based rehabilitation system to conventional therapy methods. Metrics such as hand stability, grip strength, and task performance were measured, alongside qualitative user feedback on ease of use and engagement levels. Findings from these assessments informed iterative refinements, ensuring that both the physical and virtual components align with user needs. Additionally, although the current system primarily focuses on on-site rehabilitation, healthcare professionals highlighted the potential for remote monitoring through telehealth functionalities, enabling broader accessibility and personalized long-term rehabilitation plans [7]. By synthesizing insights from structured interviews and empirical testing, the Mixed Reality-based rehabilitation system was refined to address key usability challenges, enhance adaptive feedback mechanisms, and optimize patient engagement. The integration of haptic reinforcement and immersive MR training environments demonstrates a novel, multidimensional approach to tremor rehabilitation for PD patients, offering both immediate motor stabilization and long-term therapeutic benefits.

## III. DESIGN PROCESS

### A. General Design

The development of the hand-stabilizing device followed an iterative design process aimed at optimizing ergonomics,

TABLE I
DEMOGRAPHIC PROFILE OF HEALTHCARE AND REHABILITATION PROFESSIONALS (N=6)

| Participant Group | Age | Gender | Experience (Years) | Specialization |
|---|---|---|---|---|
| Clinical Neurologist | 41 | Male | 16 | Movement Disorders, AI-assisted Rehabilitation |
| Clinical Neurologist | 53 | Female | 24 | Neurodegenerative Disease, Sensorimotor Therapy |
| Rehabilitation Therapist | 45 | Male | 18 | Upper-limb Dexterity, Assistive Devices |
| Clinical Researcher | 39 | Female | 11 | Human-Computer Interaction in Rehabilitation |
| Physical Therapist | 48 | Female | 20 | Haptic Feedback, Motion Adaptation |
| Occupational Therapist | 37 | Male | 13 | Motor Learning, AI-driven Therapy |

TABLE II
PARKINSON'S DISEASE PATIENT PARTICIPANTS: CLINICAL & REHABILITATION BACKGROUND (N=22)

| Patient ID | Age | Gender | Condition Duration (Years) | Prior Rehabilitation Experience |
|---|---|---|---|---|
| P01 | 70 | Male | 10 | No prior rehabilitation |
| P02 | 68 | Female | 9 | Virtual Reality motor training |
| P03 | 75 | Male | 12 | AI-assisted tremor therapy |
| P04 | 63 | Female | 5 | None |
| P05 | 71 | Male | 11 | Mixed Reality haptic rehabilitation |
| P06 | 66 | Female | 6 | No prior experience |
| P07 | 72 | Male | 13 | Strength-focused motor exercises |
| P08 | 77 | Female | 15 | AI-based motor adaptation therapy |
| P09 | 69 | Male | 8 | None |
| P10 | 73 | Male | 10 | Tremor reduction therapy with biofeedback |
| P11 | 61 | Female | 4 | None |
| P12 | 75 | Male | 13 | Mixed Reality-guided tremor training |
| P13 | 64 | Female | 5 | None |
| P14 | 70 | Male | 9 | AI-integrated haptic therapy |
| P15 | 79 | Female | 14 | Virtual Motor Adaptation |
| P16 | 62 | Male | 3 | None |
| P17 | 75 | Male | 12 | AI-driven VR rehabilitation |
| P18 | 71 | Female | 9 | No formal rehabilitation |
| P19 | 67 | Male | 7 | Muscle Strength Training |
| P20 | 78 | Female | 15 | None |
| P21 | 68 | Male | 8 | Adaptive hand stabilization therapy |
| P22 | 60 | Female | 2 | Mixed Reality-assisted motor training |

functionality, and user comfort. The design phase involved the creation of multiple mockups and prototypes, refining the hinge mechanism to facilitate smooth transitions between gripping and releasing while stabilizing tremors. Early design explorations focused on replicating natural hand movements, ensuring an intuitive user experience for individuals with tremor disorders. (Fig. 1).

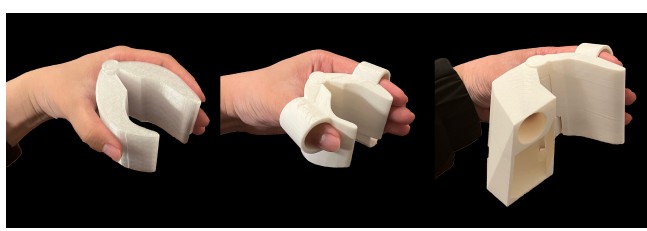

Fig. 1. Previous iterative prototypes.

The design mockups were informed by prior research on MR-assisted rehabilitation and haptic feedback integration. Studies on servo-assisted grip stabilization and real-time digital overlays demonstrated the effectiveness of combining physical support with adaptive virtual guidance [9], while research on XR tools emphasized sensor-driven feedback loops for enhanced interaction. [8] Additionally, recent findings on MR-based stroke rehabilitation highlight the benefits of real-time finger tracking and tangible user interfaces in improving motor control [10]. Drawing from these insights, our system integrates gyroscopic tracking and object recognition within a smartphone-based MR interface, ensuring seamless synchronization between physical hand movements and virtual feedback for an intuitive and adaptive rehabilitation experience.

Prototyping played a critical role in enhancing the device's form, stability, and usability. Through an ongoing cycle of development and evaluation, feedback from healthcare professionals and individuals with tremor disorders informed refinements to improve fit, comfort, and effectiveness. Particular attention was given to the hinge mechanism, ensuring that the device maintained a natural hand curvature in the resting state while enabling a seamless transition between open and closed positions. The integration of a servo motor allowed for real-time adjustments, functioning both as a movement facilitator and a stabilizer to counteract tremors and enhance grip control. The device operates through a structured control flow, as illustrated in (Fig. 2), which delineates the relationship between hand movement, servo motor response, and voice command activation. The system is structured around two primary operational states: Hand Status: Open – The hand remains in a relaxed position, allowing free movement for grasping. The servo motor remains in standby mode, following the user's natural motion. Hand Status: Closed – Upon

detecting a firm grip, the servo locks in place, stabilizing the hand and reducing involuntary tremors.

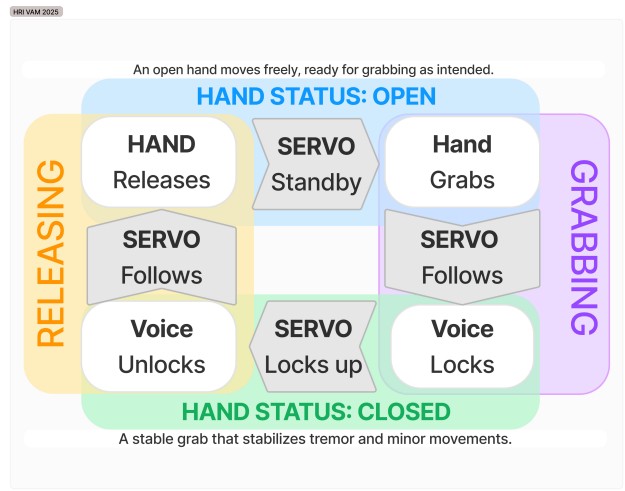

Fig. 2. Informational and functional flow chart.

The release process mirrors this structured approach. A voice command disengages the servo, returning it to a following mode and restoring the user's ability to move the hand freely. This mechanism minimizes user effort, ensuring a fluid and effortless transition between gripping and releasing. To accommodate diverse user needs, the device was designed for scalability and adaptability, allowing for variations in hand size and tremor severity. The final prototype represents a balance between stability and flexibility, offering enhanced motor control while promoting user autonomy in daily activities. By reducing muscular strain and facilitating precise, stabilized movements, the device empowers individuals with tremor disorders to perform essential tasks with greater ease and confidence.

### B. Prototype physical design

The physical prototype of the hand motion assistance device was fabricated using 3D printing technology, enabling a design that conforms precisely to the natural contours of the human hand (Fig. 3). This ergonomic structure aligns with the palm's resting curvature, ensuring stable and controlled support during hand movements. The device comprises a dual-component system connected by a hinge mechanism, replicating the opposition between the thumb and fingers, a fundamental aspect of gripping and object manipulation.

A servo motor embedded within the thumb component serves as both an actuator and a stabilizer, facilitating real-time, responsive adjustments to mitigate tremors and maintain a secure grip. Upon user-initiated grasping, the motor engages to reinforce stability, ensuring a firm hold on objects. The closed position replicates a natural firm grip, while the open position follows the palm's neutral resting state, enabling an effortless transition between gripping and releasing. A button control mechanism deactivates the servo motor to facilitate release, allowing the device to return smoothly to its resting

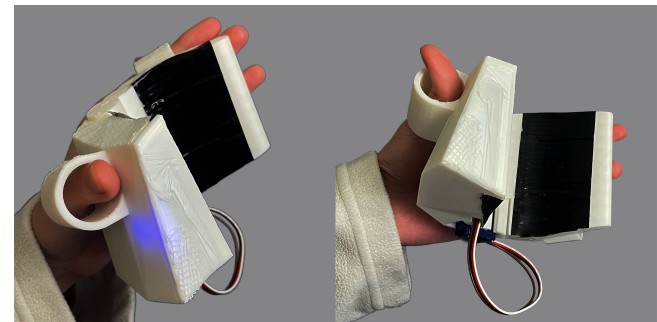

Fig. 3. Informational and functional flow chart.

position with minimal effort. This frictionless transition optimizes usability and fosters user independence in performing daily tasks. To maintain lightweight and portable functionality, the device integrates an Arduino Nano board, low-energy Bluetooth connectivity, and a compact battery housed within one handle. This wireless configuration minimizes bulk and enhances mobility while supporting potential remote monitoring and adjustment in future telehealth applications. The final prototype underwent iterative refinements informed by initial testing sessions with healthcare professionals and users, emphasizing ergonomic adaptability and mechanical precision. By aligning biomechanical design with natural hand dynamics, the prototype facilitates intuitive operation, improves dexterity, and enhances user autonomy. This technically optimized approach ensures the device is efficient, user-friendly, and tailored to the needs of individuals with tremor disorders, achieving a balance between functional support and ease of use.

### C. Mixed Reality System Development

The mixed reality (MR) system was developed to enhance rehabilitation by providing real-time interactive guidance, and improving engagement and motor skill development (Fig. 4). To maximize accessibility and affordability, the system integrates with a smartphone-based application, leveraging a Unity platform for low-cost, cross-platform compatibility. This approach eliminates the need for specialized hardware, making rehabilitation tools more widely available to individuals with movement disorders.

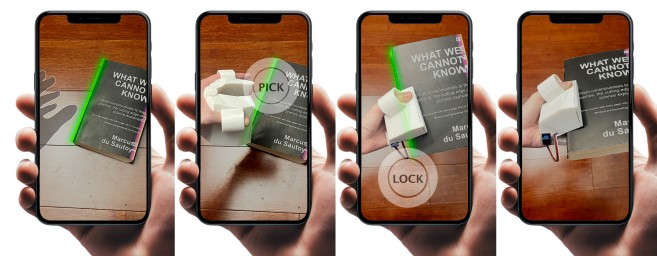

Fig. 4. Smartphone Interface of a Unity-based Mixed Reality App.

The system utilizes the smartphone's camera for object recognition, allowing users to activate the "Grab Suggestion"

feature by pointing at objects such as a book or cup. The application analyzes object orientation and size, suggesting optimal gripping techniques to improve hand positioning and grasp stability. To synchronize physical and virtual interactions, the prototype incorporates a gyroscope, which transmits real-time hand movement data to the smartphone. This enables immediate feedback on grip strength and positioning, ensuring task accuracy and bridging the gap between therapeutic exercises and functional activities. Users interact with the MR system through a structured, adaptive experience. The smartphone application provides step-by-step guidance, dynamically adjusting task difficulty as motor control improves. This progressive adaptation maintains engagement and encourages sustained participation. By integrating visual prompts with real-world interactions, the system promotes effective, intuitive learning while reinforcing rehabilitation goals. Designed for scalability, the Unity-based system supports future telehealth applications, allowing remote monitoring and personalized adjustments by healthcare providers. This capability reduces the need for frequent in-person visits while maintaining individualized rehabilitation plans. By combining real-time guidance, haptic feedback, and interactive adaptability, the MR system not only stabilizes hand movements but also fosters targeted motor training, promoting long-term functional independence.

## IV. Conclusion

This paper has presented an initial investigation into the role of mixed reality (MR) in tremor rehabilitation, particularly in the context of Parkinson's disease. By combining an ergonomic hand-support device with interactive MR-based exercises, the proposed system provides a novel framework for addressing motor instability while enhancing patient engagement through immersive and adaptive interventions. Given that PD-related tremors affect up to 80% of diagnosed individuals and significantly impair daily task execution, this study underscores the urgent need for more innovative and dynamic rehabilitation approaches. The iterative development process, guided by expert consultation and patient insights, has emphasized the usability and feasibility of MR-based rehabilitation. While the findings provide a strong foundation for future exploration, further empirical validation is necessary to assess the system's efficacy in improving motor function over extended rehabilitation periods. Future research should focus on controlled longitudinal studies, evaluating the impact of MR-assisted rehabilitation on functional motor outcomes and patient adherence rates. Beyond its immediate application in PD-related tremor rehabilitation, this study contributes to the broader discussion on MR's role in neurorehabilitation, reinforcing its potential as an effective adjunct to conventional therapy. By advancing MR-based rehabilitation methodologies, this work sets the stage for future interdisciplinary research aimed at developing scalable, technology-driven interventions that enhance patient autonomy and quality of life.

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
