# OpenReview forum: "User-Centric Mixed Reality Interventions for Parkinson’s Tremor Management: A Path Toward Digital Therapeutics"
_humanrobotinteraction.org/HRI/2025/Workshop/VAM — HRI 2025 Workshop VAM Submission_

### Official Review · Reviewer_ZQgy · 2025-02-28

**Rating:** 7
**Confidence:** 5

**Review:**

This submission presents an initial investigation into the development of a mixed reality rehabilitation system designed to support tremor rehabilitation in patients with Parkinson’s Disease (PD). The study employed a comprehensive methodology that involved interviewing 28 participants, 22 were PD consumers and 6 were healthcare professionals with experience within the field of PD. I would like to praise the authors of this work for such robust interviews with consumers with lived experience of PD and gathering critical information that provided benefit to future designs within this field. The findings from this study propose a novel framework for addressing motor instability and patient engagement through using immersive interventions, as well as prototype designs. Both outputs would provide valuable insights for the VAMHRI workshop and should be included within the event.

---

### Decision · Program_Chairs · 2025-02-26

Accept